# Evolution of ^18^F-FDG Uptake as a Pitfall of Image Diagnosis for Systemic Anaplastic Large Cell Lymphoma

**DOI:** 10.3390/diagnostics11081387

**Published:** 2021-07-31

**Authors:** Hung Chang, Wen-Yu Chuang

**Affiliations:** 1College of Medicine, Chang Gung University, Taoyuan 333423, Taiwan; chuang.taiwan@gmail.com; 2Division of Hematology, Chang Gung Memorial Hospital, Linkou 333423, Taiwan; 3Center of Hemophilia and Coagulation Medicine, Chang Gung Memorial Hospital, Linkou 333423, Taiwan; 4Division of Hematology-Oncology, Chang Gung Memorial Hospital, Linkou 333423, Taiwan; 5Department of Pathology, Chang Gung Memorial Hospital, Linkou 333423, Taiwan

**Keywords:** anaplastic large cell lymphoma, PET, CT scan

## Abstract

In most patients, systemic anaplastic large cell lymphoma (sALCL) is an ^18^F-FDG-avid tumor. Both ALK-positive and ALK-negative tumors can be evaluated by PET scans as both tumor types uptake ^18^F-FDG in PET. The PET scan is also valuable in predicting prognosis during and after the treatment course. The evolution of ^18^F-FDG uptake in patients with sALCL has not been reported. For tumors lacking ^18^F-FDG uptake, there is a diagnostic pitfall of underestimating the cancer stage and misjudgment of metastases. In the present case, the PET scan results were negative at diagnosis but disseminated ^18^F-FDG avid lesions were found at relapse. Biopsy of bone marrow and lymph nodes revealed the pathological features were identical to the original tumor at the time of diagnosis. In the wake of such evolutional change, physicians dealing with sALCL should be cautious in interpretation of PET/CT scans.

Systemic anaplastic large cell lymphoma (sALCL) is an aggressive T cell lymphoma. ALK-positive and ALK-negative sALCL have distinctive clinical features and prognosis [1,2]. In general, sALCL is an ^18^F-FDG avid tumor although, the SUV is different between ALK-positive and ALK-negative sALCL [3,4]. As most tumors of sALCL are ^18^F-FDG avid [3,4], a PET scan is recommended in defining tumor stage and evaluating treatment response [4]. In particular, the PET scan is useful in identifying additional sites of disease in peripheral T cell lymphomas, including sALCL [5]. In addition, interim and post-treatment PET scans are useful in predicting treatment outcome [6].

In the present case, tumors were negative for ^18^F-FDG uptake at diagnosis. The reason for such lack of uptake may be tumor necrosis, as revealed by pathological examinations. PET scans may show tumor necrosis as photopenic defects, and therefore affect interpretation of viable tumors. However, at relapse, all lesions were ^18^F-FDG avid, and biopsies confirmed the tumors were identical to previous lymphoma at diagnosis except for lack of necrosis. Evolution of ^18^F-FDG uptake can be well demonstrated in such an experience. In view of such evolution, especially the negative ^18^F-FDG uptake at diagnosis, we suggest in sALCL, PET scans should be interpreted with caution. Misinterpretation may result in underestimates of tumor stage and misdiagnosis of relapse (Figure 1).

## Figures and Tables

**Figure 1 diagnostics-11-01387-f001:**
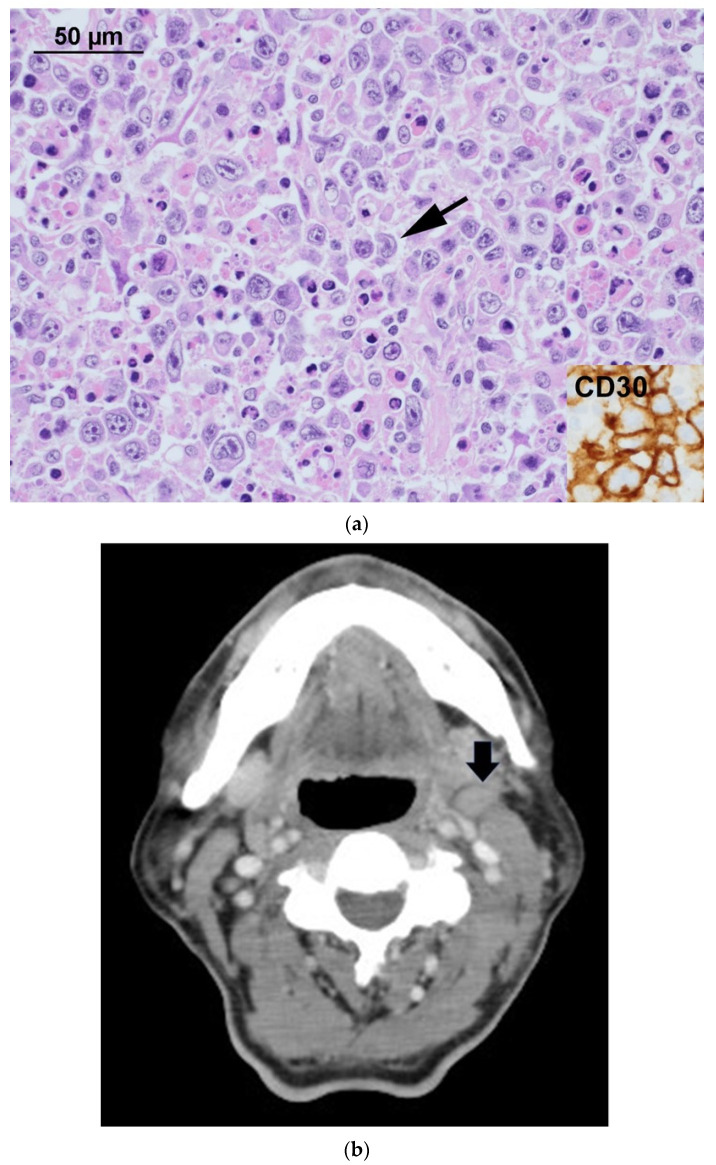
A 63-year-old man presented with left neck swelling. A core needle biopsy was performed. Microscopically, there were diffuse infiltrates of large lymphoid cells with pleomorphic nuclei, prominent nucleoli and abundant cytoplasm. Occasional tumor cells with horseshoe-shaped nuclei (arrows) could be found. The tumor cells were strongly positive for CD30 (insets). They were also positive for CD2, CD4, CD8 and TIA-1. The tumor cells were negative for CD20, PAX5, CD3, CD56 and ALK. The primary tumor had prominent tumor necrosis, which was absent in the recurrent tumor. The diagnosis of ALK-negative sALCL was made (**a**). An ^18^F-FDG PET/CT scan was carried out with the Siemens Biograph mCT PET/CT scanner (370 MBq (10mCi), 18F-FDG injected and patient scanned at 50 min from the mid-thigh to the skull vertex). Although enlarged lymph nodes in the left neck area were found in the CT part of PET/CT scan (**b**), there was only minimal ^18^F-FDG uptake (maximal SUV 2.54), considered to be negative for lymphoma in that area (**c**). The patient underwent CHOP (cyclophosphamide, doxorubicin, vincristine, prednisolone) chemotherapy for 4 cycles and had a complete response according to the interim CT scan. A total of six cycles were administered. However, one month following completion of chemotherapies, the patient reported fever and enlarged lymph nodes in the right neck. Another PET/CT scan showed increased ^18^F-FDG uptake in cervical, mediastinal, mesenteric, retroperitoneal, iliac regions, spleen and skeletal bones (**d**). Bone marrow and neck lymph node biopsies showed relapse of ALK-negative sALCL (**e**).

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
