# Peer review of "Evolution of ^18^F-FDG Uptake as a Pitfall of Image Diagnosis for Systemic Anaplastic Large Cell Lymphoma"

_diagnostics, 2021, doi:10.3390/diagnostics11081387_

Round 1

Reviewer 1 Report

The Authors present an unusual case of a patient with anapaestic large cell lymphoma with a negative PET/Ct at diagnosis and positive at relapse. The Authors suggest that a PET/CT in anapaestic large cell lymphoma should be interpreted with caution. This is true since PET/CT is always an additional test and always should be interpret with caution also in case of unusual negative result. The paper lacks a proper discussion of possible explanation of negativity of baseline PET/CT since it is very unlikely that indeed PET/CT was negative ans positive at the time of relapse 

Reviewer 2 Report

The case report in interesting and well done. The figure are very beautiful and the topic investigated is original.

Only some minor points to improve:

  • the "pathological" images are less clear for a non pathologist. Can you explain better there images?
  • in Figure C, in the MIP I see a focla uptake in left neck. Node? aspecific? Please clarify
  • Nothing about PET/CT scanner and protocol i sdescribed? shich type of scanner? digital? .....
  • In figure B, do you mean CT of PET or a diagnostic CT? Please clarify

Round 2

Reviewer 1 Report

After additional information provided by the AUthors, the report can be accepted for publication